# The Incidence of Scabies in Far North Queensland, Tropical Australia: Implications for Local Clinical Practice and Public Health Strategies

**DOI:** 10.3390/tropicalmed10040111

**Published:** 2025-04-18

**Authors:** Mert Hamdi Korkusuz, Maria Eugenia Castellanos, Linton R. Harriss, Allison Hempenstall, Simon Smith, Josh Hanson

**Affiliations:** 1Cairns and Hinterland Hospital and Health Service, Cairns, QLD 4870, Australia; mhkorkusuz@gmail.com (M.H.K.); linton.harriss@health.qld.gov.au (L.R.H.); allison.hempenstall@health.qld.gov.au (A.H.); simon.smith2@health.qld.gov.au (S.S.); 2College of Public Health, Medical and Veterinary Sciences, James Cook University, Townsville, QLD 4811, Australia; maria.castellanosreynosa@jcu.edu.au; 3College of Public Health, Medical and Veterinary Sciences, James Cook University, Cairns, QLD 4870, Australia; 4Kirby Institute, University of New South Wales, Sydney, NSW 2052, Australia

**Keywords:** scabies, acute rheumatic fever, rheumatic heart disease, chronic kidney disease, social determinants of health, aboriginal and Torres Strait Islander Australian health, tropical medicine

## Abstract

The recognition and treatment of scabies has been incorporated into Australian guidelines for the prevention of acute rheumatic fever (ARF) and rheumatic heart disease (RHD). The incidence of both diagnosed ARF and RHD is increasing in Far North Queensland (FNQ) in northeast tropical Australia, but the local burden of scabies is incompletely defined. We reviewed the results of every skin scraping collected in FNQ’s public health system between 2000 and 2023; 121/4345 (2.8%) scrapings were positive, including 19/1071 (1.8%) in the last 5 years of the study; the proportion of scrapings that were positive for scabies declined over the study period. Individuals who tested positive for scabies were no more likely to have had a prior diagnosis of ARF or RHD compared to the matched controls (1/101 (1%) versus 3/101 (3%), *p* = 1.0). During a median of 14.7 years of follow-up, individuals who tested positive for scabies were also no more likely to have a diagnosis of ARF or RHD than matched controls (2/100 (2%) versus 6/98 (6%); hazard ratio (95% confidence interval): 0.30 (0.06–1.50) *p* = 0.14). Microbiologically confirmed scabies is uncommon in FNQ and appears to make a limited contribution to the local incidence of ARF and RHD.

## 1. Introduction

Scabies, an ectoparasitic skin infestation caused by the mite *Sarcoptes scabiei var hominis*, affects more than 200 million people globally [1]. It primarily occurs in disadvantaged populations, is more common in tropical regions, and was defined as a Neglected Tropical Disease by the World Health Organisation in 2017 [2,3,4]. In Australia, scabies disproportionately affects Aboriginal communities in the tropical north of the country [2,5].

Infestation with scabies mites results in a pruritic rash that can profoundly affect quality of life. The rash can impair sleep, affect an individual’s ability to work, and lead to stigmatisation and embarrassment [6,7,8]. Scabies is also predisposed to bacterial superinfection, often by *Staphylococcus aureus* or *Streptococcus pyogenes,* which can result in life-threatening sepsis [5,9,10]. Additional sequelae of *S. pyogenes* infection include acute rheumatic fever (ARF) and rheumatic heart disease (RHD), and post-streptococcal glomerulonephritis (PSGN) [11,12,13].

ARF is an immune-mediated inflammatory condition that follows *S. pyogenes* infection and that can involve the skin, subcutaneous tissue, joints, brain, and the heart [14,15]. Severe or recurrent episodes of ARF-induced carditis can result in permanent cardiac damage, a condition termed RHD. In 2019, it was estimated that 40.5 million people worldwide were living with RHD and that it caused almost 310,000 deaths [16]. The rates of ARF and RHD in Aboriginal and Torres Strait Islander Australians (hereafter respectfully referred to, collectively, as First Nations Australians) are among the highest recorded in the world [17]. *S. pyogenes* infection is also associated with PSGN, which is increasingly recognised as a risk factor for chronic kidney disease (CKD) [18,19]. Globally, PSGN is the most common cause of acute nephritis in children, with more than 95% of cases occurring in developing countries, although again, the incidence of PSGN is very high in many First Nations Australian communities [13]. RHD and CKD are both important causes of premature morbidity and mortality in First Nations Australians in tropical Australia [17,20].

In classic scabies, the most common manifestation of scabies, individuals have fewer than 20 mites, and the most prominent symptom is pruritus [21]. A severe form of scabies, crusted scabies, is characterised by hyper-infestation with millions of mites and the formation of hyperkeratotic crusts [22]. Crusted scabies is associated with a much higher rate of complications than classic scabies, is far more contagious, and therefore has significant public health implications [23,24].

Early recognition and treatment of scabies improves quality of life and would also be expected to reduce the incidence of secondary bacterial skin infections and their sequelae. Indeed, scabies recognition and treatment has been incorporated into guidelines and programmes for the management of ARF/RHD and PSGN [25,26,27,28]. Identifying and treating scabies is an attractive public health intervention, as treating individual cases is relatively simple. Both topical (permethrin) and oral (ivermectin) medications are available, effective, and generally well tolerated [29]. However, due to the challenges of timely treatment of household contacts, and the resulting high rates of reinfection, scabies remains endemic in many First Nations communities in tropical Australia [1,30].

The majority of Australian research into scabies occurs in the Northern Territory (NT), where guidelines for both individual cases and public health strategies have been developed [31,32,33]. There is interest in applying these strategies elsewhere in Australia, particularly in First Nations communities. One of these regions is FNQ in the tropical northeast of the country, where 17% of the population identifies as First Nations Australians [34]. The FNQ region has a significant burden of skin infection, ARF/RHD, and CKD, and has three of the seven most disadvantaged Local Government Areas in Australia [35,36,37,38,39]. Given the similar demographics, similar climate, and similar impact of the social determinants of health, it might be expected that there is a similar incidence of scabies and crusted scabies in FNQ to that seen in the NT.

However, the anecdotal experience of physicians who have worked across Northern Australia is that the prevalence and severity of scabies appears to be lower in FNQ than the NT. If true, some interventions proposed in the NT, such as mass administration of ivermectin, may have a limited impact in FNQ. Public health strategies that address access to care, health literacy, and housing adequacy may represent higher-value interventions to reduce the local burden of ARF/RHD and CKD [17,20,40,41].

Therefore, this study was conducted to define the temporospatial epidemiology of scabies infestation in FNQ. It also evaluated whether individuals with confirmed scabies had a higher risk of subsequently developing ARF, RHD, and renal impairment. Finally, it examined whether the practice of local clinicians assessing individuals at risk of scabies was in accordance with contemporary published guidelines. It was hoped that these data would help inform strategies to improve the skin health of people living in the FNQ region, reduce the local incidence of RHD and PSGN, and optimise the use of finite health resources.

## 2. Methods

### 2.1. Study Design and Population

This retrospective study was performed at Cairns Hospital, the 581-bed tertiary referral hospital for the FNQ region of tropical Australia (Figure 1). The hospital serves a population of approximately 290,000 people who live across an area of 380,000 km^2^ [34]. There are two health services in the region, the Cairns and Hinterland Hospital and Health Service (CHHHS)—which serves the population of the administrative hub of Cairns and the surrounding region—and the Torres and Cape Hospital and Health Service (TCHHS), which serves the almost exclusively rural and remote population living on the Cape York Peninsula and the Torres Strait Islands.

### 2.2. Identification of Individuals with a Positive Skin Scraping in the FNQ Region Between 2000 and 2023

The Cairns Hospital laboratory is the referral laboratory for all microbiology services in the public health system in the FNQ region and the sole facility that performs microscopy of skin scrapings. All laboratory results collected in the state of Queensland’s public health system are entered into the electronic database AUSLAB. The AUSLAB database was interrogated to identify all skin scraping requests processed in the Cairns Hospital laboratory between 1 January 2000 and 31 December 2023.

The patients’ demographics and the date and location of the request were recorded. Tests that were not performed on skin were removed, as were tests where microscopy could not be performed. Duplicate tests (performed on the same individual within 30 days) were also removed; if any duplicate tests within the 30-day period were positive for scabies mites, eggs, or faecal pellets, this was defined as a positive episode. If all of these duplicate tests were negative, this was defined as a negative episode. All individuals receiving care in Queensland’s public health system are asked whether they identify as First Nations Australian. Individuals aged 15 years and younger were defined as children. Individuals tested in Cairns were said to have had testing in an urban location; those tested outside of Cairns were said to have had testing in a rural or remote location.

### 2.3. The Association Between a Positive Skin Scraping and Subsequent Diagnosis of ARF/RHD or CKD

A list of the individuals that had a positive skin scraping for scabies between 2000 and 2023 was generated, and a corresponding list of controls (matched to age, sex, location, and First Nations Australian status) who tested negative was selected manually. The date that the scraping was performed was also matched (+/−2 years) to ensure that there was a similar duration of follow-up. ARF and RHD have been notifiable diseases in the state of Queensland since 1998 and 2018, respectively. The Queensland Rheumatic Heart Disease register was reviewed to determine if these individuals had been notified to the register for either ARF or RHD, and, if they had been, the date of their first notification. The individuals’ most recent formal estimated glomerular filtration rate (eGFR) was also identified through the AUSLAB system. CKD was defined as an eGFR < 60 mL/min/1.73 m^2^ [43]. Patients who had a diagnosis of ARF or RHD or a diagnosis of CKD preceding their skin scraping were excluded from the respective analyses.

### 2.4. Assessment of Individuals Presenting to the Cairns Hospital Emergency Department with a Diagnosis of Scabies

Patients presenting to the Cairns Hospital Emergency Department who had a possible diagnosis of scabies were identified using the Emergency Department’s electronic health record system, FirstNet (Oracle Health, version 19.19, Microsoft, Kansas City, MO, USA). This system was established in Cairns Hospital in March 2016. All presentations that had the word “scabies” recorded in the electronic medical record between 1 March 2016 and 31 May 2024 were identified; re-presentations within 30 days were excluded. Each patient was cross-referenced with AUSLAB to determine if skin scrapings were collected in the public health system 30 days before or after their presentation. The patients’ medical records were reviewed, and demographic and relevant clinical data were recorded. The diagnostic approach of the attending clinicians was recorded to determine if this was in accordance with IACS guidelines (Appendix A), and if the assessment of possible cases of crusted scabies followed NT guidelines [32].

### 2.5. Statistical Analysis

Data were de-identified, entered into an electronic database (Microsoft Excel version 16.0, Microsoft, Redmond, WA, USA), and analysed using statistical software (Stata version 18.0, StataCorp LLC., College Station, TX, USA). Trends over time were analysed using Spearman’s test for correlation or the Cochran–Armitage test. Groups were compared using logistic regression, the chi-squared test, McNemar’s test, or Fisher’s exact test, as appropriate. Multivariate analysis was performed using logistic regression with variables selected for the multivariate model if their *p*-value in univariate analyses was <0.10. The subsequent incidence of ARF/RHD or an eGFR < 60 mL/min/1.73 m^2^ in individuals with positive skin scrapings and matched controls were analysed using a Cox proportional hazards model and presented using Kaplan–Meier curves. Individuals with missing data were not included in analyses that evaluated those variables.

### 2.6. Ethics Statement

The study was conducted in accordance with the Declaration of Helsinki, and approved by the Far North Queensland Human Research Ethics Committee (EX/2023/QCH/102367 (Oct ver 3) -1766 QA) on the 27 October 2023. As the retrospective data were de-identified and presented in an aggregated manner, the Committee waived the requirement for informed consent.

## 3. Results

### 3.1. Skin Scrapings

A review of the AUSLAB electronic laboratory database identified 6199 episodes that were recorded as skin scrapings. After excluding duplicates and non-skin samples, a total of 4345 distinct testing episodes were included in the analysis (Appendix A).

### 3.2. Characteristics of the Individuals Who Were Tested

From 2000 to 2023, there was a median of 188 annual skin scraping episodes (interquartile range (IQR): 162–224). The number of annual episodes increased during the study period (r_s_ = 0.64, *p* = 0.001) (Figure 2). The median age of patients with skin scrapings taken was 42 years (IQR 23–58, range 0–100), and 790/4345 (18%) were children. Of the 4342 episodes where the location of testing was available, 3397 (78%) scrapings were collected in rural or remote locations. Of the 4314 episodes where First Nations Australian status was available, the individuals identified as a First Nations Australian in 2707 (63%).

### 3.3. Number of Positive Tests and Trend over Time

Scabies was diagnosed in 121/4345 (2.8%) skin scraping episodes. The 121 diagnoses of scabies occurred in 101 individuals. The proportion of tests that were positive for scabies declined over the study period (*p* = 0.0001) (Figure 3). Indeed, scabies was diagnosed in only 19/1071 (1.8%) scrapings in the last 5 years of the study. In the multivariate analysis, positive tests were more common in First Nations Australians, in adults, and in individuals tested in an urban location (Table 1).

### 3.4. Association of Microbiologically Diagnosed Scabies with a Subsequent Diagnosis of ARF/RHD or CKD

Individuals who tested positive for scabies were no more likely to have had a prior diagnosis of ARF or RHD compared to the matched controls (1/101 (1%) versus 3/101 (3%); *p* = 1.0). Furthermore, during a median (IQR) of 14.7 (6.6–20.1) years of follow-up, individuals who tested positive for scabies were also no more likely to have a diagnosis of ARF or RHD than the matched controls (2/100 (2%) versus 6/98 (6%); hazard ratio (HR) (95% CI): 0.30 (0.06–1.50) *p* = 0.14) (Figure 4 and Appendix A).

There were 168/202 (83%) individuals with skin scrapings who had an available eGFR; in 154/168 (92%) individuals, this was greater than 30 days after the skin scraping. Individuals who tested positive for scabies were not at an increased risk of having a GFR < 60 mL/min/1.73 m^2^ during follow-up than the matched controls (43/70 (61%) versus 40/84 (48%); HR (95% CI) 1.39 (0.90–2.14), *p* = 0.13) (Figure 5 and Appendix A).

### 3.5. Assessment of Individuals Presenting to the Cairns Hospital Emergency Department with Clinically Suspected Scabies

There were 353 presentations to Cairns Hospital Emergency Department coded as a presentation of scabies. After re-presentations within 30 days, and individuals who were subsequently confirmed to have an alternative diagnosis were removed, 292 presentations in 265 individuals were included in this analysis; 21/292 (7.2%) had a clinical suspicion of crusted scabies.

### 3.6. Classic Scabies

Only 9/271 (3.3%) with clinical suspicion of classic scabies had skin scrapings performed, none of which were positive for scabies (Table 2 and Appendix A). Only 86/271 (31.7%) were reviewed by a specialist of any specialty (including the specialty of emergency medicine) during their hospital encounter. There was no documented use of dermatoscopy or high-powered imaging in any of the cases. The medical records of only 11/271 (4.1%) contained documentation suggesting that the clinicians involved had addressed all the IACS criteria for a diagnosis of scabies (Appendix A). However, recognising that not all IACS criteria need to be addressed in order to satisfy the diagnostic criteria for “clinical scabies”, 88/271 (32.4%) cases had sufficient recorded data to support the diagnosis. There was no discernible change in the documentation of clinical findings, nor the performance of skin scrapings, after March 2020, when the IACS criteria for the diagnosis of scabies were published [44].

### 3.7. Crusted Scabies

Only 5/21 (23.8%) cases where crusted scabies was considered had skin scrapings collected, of which only 1/5 (20%) was positive (Table 2 and Appendix A). The medical records of only 4/21 (19%) cases contained documentation to suggest that the clinicians had addressed all aspects of the IACS criteria for the diagnosis of scabies (Appendix A). Overall, there were 13/21 (61.9%) cases of suspected crusted scabies that satisfied the IACS criteria for a diagnosis of clinical scabies. Only 14/21 (66.7%) were reviewed by a specialist of any specialty (including the specialty of emergency medicine) during their hospital encounter, and only 6/21 (28.6%) had review by the dermatology or infectious diseases services; two of these (both seen by dermatology) were the only two cases that had a severity grading assessment, and one of these cases (seen by dermatology) was the only case that had dermatoscopy documented [46]. There was only 1/21 (4.7%) suspected case that satisfied the NT guidelines criteria for confirmed crusted scabies. There was only one case (not the confirmed case) that had a complete initial evaluation in accordance with the NT guidelines.

Only 16/21 (76%) individuals with suspected crusted scabies were admitted to hospital, and their hospitalisations were only for a median (IQR) of 3 (3–5) days. Only 16/21 (76%) received any ivermectin, although this was sometimes in the setting of concerns about the age of a paediatric patient. The median (IQR) number of ivermectin doses was 5 (3–5), but only 3/21 individuals received keratinolytic therapy, and documentation about environmental control and treatment of family members was present in only 9/21 (43%) and 10/21 (48%), respectively.

## 4. Discussion

This study of scabies in the FNQ region of tropical Australia had three main findings. The first was that less than 3% of skin scrapings within the FNQ public health system between 2000 and 2023 were positive for scabies and that the proportion of positive tests declined over this period. Indeed, despite an increasing annual collection of skin scrapings, there were fewer than 20 microbiologically confirmed cases of scabies in >1000 episodes of testing in the last five years of the study. Secondly, despite a significant local burden of ARF/RHD and CKD, the individuals who tested positive for scabies were no more likely to subsequently develop ARF/RHD or renal impairment than matched controls who tested negative, suggesting a limited contribution of scabies infection to the development of these complications in the FNQ region. Thirdly, the diagnostic assessment of individuals presenting to the region’s tertiary referral hospital with a possible diagnosis of scabies was frequently suboptimal, particularly in those with a possible diagnosis of crusted scabies. Clinical suspicion was rarely confirmed microbiologically, even when there was concern for crusted scabies. Other elements of the recommended clinical assessment and management of a patient with suspected scabies were also documented poorly, suggesting the need for a reassessment of current clinical practice in the Cairns Hospital Emergency Department, and, potentially, the FNQ region more broadly. Greater adherence to contemporary diagnostic approaches will help define the true burden of scabies in the region and the implications for local clinical practice and public health strategies.

First Nations Australians accounted for over 90% of the microbiologically confirmed cases of scabies in the region, despite accounting for only 17% of the local population, which is consistent with the epidemiology of scabies reported from other Australian jurisdictions [2,5]. However, in contrast to other Australian studies, a confirmed diagnosis of scabies was, in multivariate analysis, over six times less likely in rural and remote communities in this study. This does not appear to be wholly explained by lower testing rates in rural and remote locations, as more than half of the skin scrapings were collected in remote communities in the Torres Strait and Cape York regions, where less than 10% of FNQ’s population resides. A microbiologically confirmed diagnosis of scabies was also almost six times less common in children. This was surprising, as scabies is consistently reported in the literature to be more prevalent in children, to the extent that monitoring scabies in children has been proposed as the most effective method to track prevalence and detect outbreaks [2,4,31,47]. While this finding might be partly explained by clinicians’ reluctance to perform potentially distressing skin scrapings in children, over 18% of the skin scrapings in our cohort were performed in individuals aged <16 years of age (less than 1% of which were positive, compared to >3% in adults), while 13% of the FNQ population is aged <20 years of age [34]. The fact that skin scrapings were performed in a variety of geographic settings and across the full age-range suggests that there are no systemic barriers to performing this test.

Instead, it appears that local clinicians have a greater inclination to diagnose scabies clinically. This is supported by the finding that less than 5% of individuals seen in the emergency department with suspected scabies—and less than a quarter with suspected crusted scabies—had skin scrapings collected. There was also only a single case where the use of high-powered imaging or dermatoscopy was documented. The availability, efficacy, and ease of use of ivermectin likely reduces the impetus for busy clinicians to perform confirmatory testing [48]. But without confirmatory testing, there is a risk of bias informing the clinical diagnosis of scabies, particularly in populations believed to be at greater risk [49]. The clinical diagnosis of scabies is challenging, relies heavily on the experience of the clinician, and its accuracy is difficult to verify [50,51]. Even after dedicated training, healthcare professionals have difficulty in identifying cases [52]. Many of the individuals with suspected scabies in this cohort were seen only by junior clinicians—with little to no training in the diagnosis of scabies—casting some doubt as to the accuracy of their clinical assessment. Although the documentation was often incomplete and the medical charts were reviewed retrospectively, only about 35% of the individuals in our cohort with a diagnosis of scabies in the Emergency Department satisfied IACS criteria for a diagnosis of clinical scabies. While scabies is under-recognised globally, overdiagnosis and overtreatment are also seen in endemic or presumed-endemic regions [53,54]. The limited documentation of the clinicians’ assessment seen in this cohort may reflect documentation practices in a busy emergency department, a manifestation of the recognised phenomenon of the trivialisation of skin conditions, or further evidence of limited familiarity with scabies among FNQ health workers [53,55].

The very low rate of confirmed scabies in FNQ is consistent with anecdotal observations of local clinicians who have worked across tropical Australia, who believe there may be a lower burden of scabies in FNQ. However, there are data that challenge this view. A public health response to a 2021 outbreak of PSGN in Aurukun, a remote Aboriginal community in Cape York, identified a scabies rate of 21% of the total population screened, and of 26% in the subgroup of screened children, rates that are comparable to the high rates seen in Top End Communities in the NT [2,5,47,56]. However, none of the diagnoses were confirmed with laboratory testing, and the staff conducting clinical assessments had limited experience and training in diagnosing scabies (Neville, J 2023, oral communication, 19 May 2023 [57]). The association with a PSGN outbreak may have increased the risk of confirmation bias [49].

Our study does provide data that suggest that crusted scabies is far less common in FNQ than other parts of tropical Australia. Crusted scabies has been a notifiable disease in the NT since 2016, and notified cases in that jurisdiction must satisfy both strict laboratory and clinical criteria, including positive skin scrapings and confirmation by either a dermatologist or infectious diseases specialist. There were 210 cases of crusted scabies between March 2016 and July 2023 in the NT’s Top End Region, which has a population of approximately 200,000 [33]. Crusted scabies is not a notifiable condition in QLD, but during the same March 2016 to July 2023 period, there were only 31 laboratory confirmed cases of any—not just crusted—scabies in the FNQ public health system, which serves a population of approximately 290,000 [58].

Of course, this may again be at least partly explained by inadequate testing, and it is concerning that over three-quarters of the individuals with a clinical diagnosis of crusted scabies presenting to the Cairns Hospital Emergency Department did not have confirmatory laboratory testing. Laboratory confirmation of crusted scabies is important, as the condition is associated with a higher rate of complications—including life-threatening sepsis—and up to 25% of individuals diagnosed with crusted scabies will die within 12 months [24,45,59]. A diagnosis of crusted scabies also has significant public health implications due to its contagiousness, with patients with crusted scabies having been identified as the index case in outbreaks [30,60]. Conversely, the appearance of crusted scabies can also be non-specific and may mimic other diagnoses, some of which can be life-threatening and require very different management [44]. Millions of mites may be present in a patient with crusted scabies, so correctly collected skin scrapings effectively exclude the diagnosis if they are negative [22,61]. NT guidelines also recommend specialist dermatology or infectious disease involvement, an assessment of severity, and admission to hospital [45]. However, a minority of the cases in this FNQ series had specialist review, very few had a complete assessment documented and almost a quarter were not hospitalised. Those who were rarely received keratinolytic therapy, and only a minority had documentation about environmental control or treatment of family members. This suggests that FNQ clinicians are unfamiliar with the recommended management of a patient with possible crusted scabies, which may, in turn, be at least partially explained by a lower incidence of crusted scabies in the region.

It has been suggested that earlier identification and treatment of scabies will reduce the risk of subsequent infection with *S. pyogenes* and therefore reduce the risk of ARF/RHD and CKD [3]. However, our study provides little evidence to support the suggestion that scabies contributes significantly to the development of either ARF/RHD or CKD in the FNQ region. The incidence of laboratory-confirmed scabies was very low in our cohort, and there was no significant difference in the development of CKD or ARF/RHD in FNQ between laboratory-confirmed cases and matched controls. It is likely that the high prevalence of CKD in the FNQ region is more strongly linked to traditional risk factors, particularly diabetes [62,63,64]. While ARF and RHD, like scabies, are strongly linked to the social determinants of health, the FNQ region, with the highest prevalence of RHD, had only two confirmed cases of scabies in the 24 years of our study [36,65].

Our study systematically evaluates laboratory data collected over 24 years, and over 8 years of hospital presentation data to provide a temporospatial estimate of the burden of scabies in the FNQ region. The study also allows for a critical appraisal of local clinical practice. However, it also has many limitations. Our finding that skin scrapings were performed in fewer than 5% of presentations with a diagnosis of scabies may also, almost certainly, underestimate the local disease incidence. The sensitivity of skin scrapings in cases of non-crusted scabies has been reported to be less than 50%, and the collection of adequate scrapings is operator dependent [1,44,66,67]. However, these factors need to be balanced against the fact that there were less than 20 cases of confirmed scabies across the public health system across the whole FNQ region in the last 5 years of the study. Only scrapings collected in the public health system were reviewed, although testing for scabies is performed infrequently in private laboratories in the region and is unlikely to have had a significant impact on the overall results or conclusions of this study. It was not possible to determine the indication for the skin scrapings, and the low and declining proportion of cases may be at least partially explained by an increasing number of scrapings being conducted for other indications—for instance, dermatophyte infection—although these conditions are also often treated empirically. A very low incidence of crusted scabies does not necessarily mean a lower incidence of all scabies infection: programmes that have conducted whole population screening have reported on the absence of even a single case of crusted scabies, despite an overall scabies prevalence of up to 18.7% [68,69]. There are well-recognised inherent limitations in using clinical coding to identify patients with specific health conditions [70,71]. On review of the medical records, it was found that over 10% of the presentations to the Cairns Hospital Emergency coded as scabies had another diagnosis confirmed or simply no evidence of scabies (Appendix A). The study’s retrospective design and reliance on medical record documentation precluded the collection of comprehensive clinical data, and almost a quarter of the individuals did not have an accessible measure of glomerular filtration rate during follow-up. For scabies and ARF/RHD to be diagnosed, the individuals need to have symptoms that lead them to present for—and receive—specific assessment. Many individuals who are diagnosed with RHD in middle age have no recollection or documented history of ARF, which may mean that we underestimated the rate of subsequent ARF/RHD in the cohort [36]. The evaluation of the association between a laboratory diagnosis of scabies and the subsequent development of ARF/RHD or CKD did not consider the contribution of potential confounding factors.

Acknowledging these limitations and recognising that it is likely that there is both under- and overdiagnosis of scabies in FNQ, our data support experienced clinicians’ impression that the local incidence of scabies is far lower than in other regions of tropical Australia. Future prospective studies that utilise standardised clinical assessment, as well as routine use of skin scrapings, dermatoscopy, and evolving molecular techniques, will better quantify the prevalence of scabies across FNQ and the implications for local public health strategies [44,72,73]. This will also facilitate optimal management of patients with both scabies and other dermatological conditions [74,75]. Making crusted scabies a notifiable condition in Queensland, as it is in the NT, would also ensure appropriate patient evaluation and management, as well as more reliable tracking of cases, reducing the risk of ongoing transmission and minimising the likelihood of outbreaks [30,60].

## 5. Conclusions

Microbiologically confirmed scabies infection is rare in the FNQ region, and despite an increase in local testing, the incidence of microbiologically confirmed cases is declining. FNQ individuals who have positive skin scrapings for scabies are no more likely to subsequently develop ARF/RHD or CKD than those with negative tests. The incidence of crusted scabies in the FNQ appears to be considerably lower than in other regions in tropical Australia. Most cases of scabies in the region are diagnosed clinically, even when crusted disease is suspected, although the evaluation of patients frequently does not follow contemporary guidelines for disease assessment and management.

Despite this, there are limited data to suggest that there is a major burden of scabies in the region. The social determinants of health have an important impact on the incidence of communicable and non-communicable diseases in FNQ and, importantly, also increase the risk of scabies emerging as an important local health issue [76,77]. While scabies recognition and treatment has been incorporated into guidelines and programmes for the management of ARF/RHD and PSGN in other parts of Australia and mass administration of ivermectin has been implemented in some communities [25,26,27,28,31], in FNQ, a greater focus on strategies that address the ongoing socioeconomic disadvantage in the region is likely to be a better use of finite local health resources.

## Figures and Tables

**Figure 1 tropicalmed-10-00111-f001:**
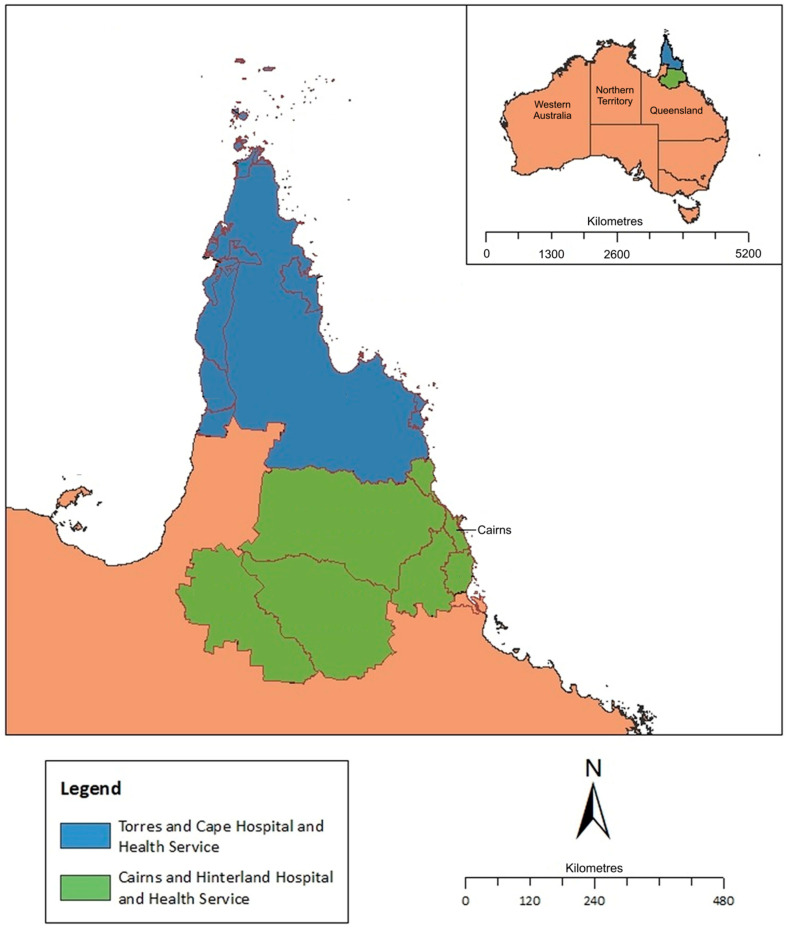
Map of tropical Far North Queensland, Australia, showing catchment area for the current scabies study (1 January 2000 to 31 December 2023). Image adapted from Bird, K. et al. [42].

**Figure 2 tropicalmed-10-00111-f002:**
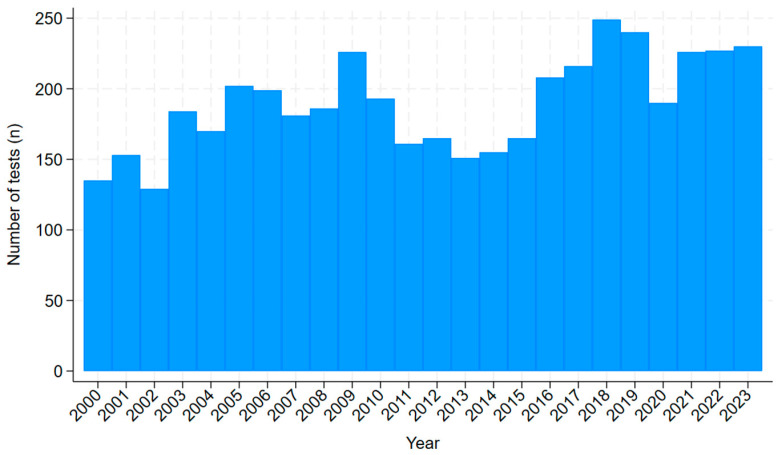
Number of annual skin scraping episodes collected over a 24-year period (1 January 2000 to 31 December 2023) in the public health system in tropical Far North Queensland, Australia.

**Figure 3 tropicalmed-10-00111-f003:**
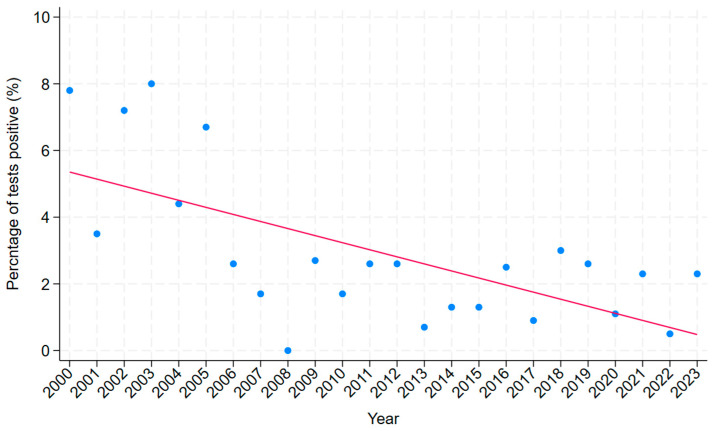
Percentage of skin scraping episodes that were positive for scabies collected over a 24-year period (1 January 2000 and 31 December 2023) in the public health system in tropical Far North Queensland, Australia (with line of best fit to highlight trend).

**Figure 4 tropicalmed-10-00111-f004:**
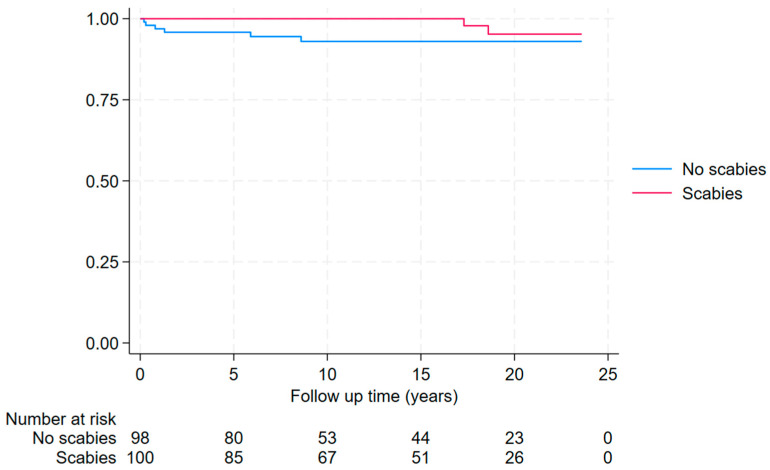
Kaplan–Meier curve showing time to new diagnosis of ARF or RHD after the episode of skin scraping between 2000 and 2023 in Far North Queensland (individuals with ARF or RHD prior to skin scraping excluded from this analysis).

**Figure 5 tropicalmed-10-00111-f005:**
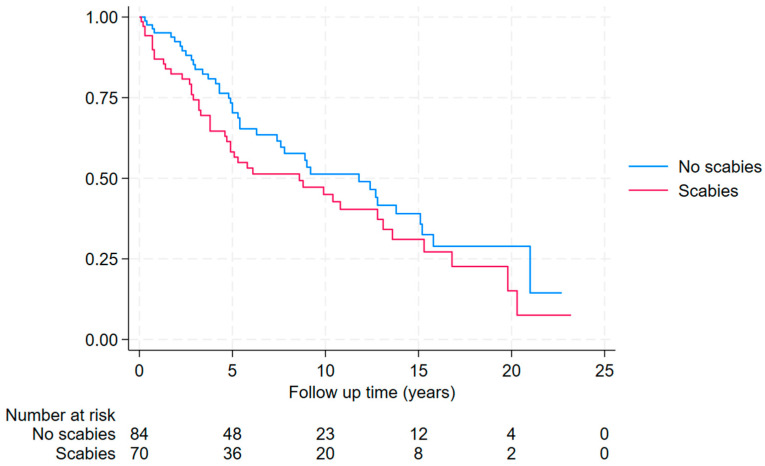
Kaplan–Meier curve showing time to eGFR < 60 mL/min/1.73 m^2^ during follow-up after the episode of skin scraping between 2000 and 2023 in Far North Queensland (individuals with eGFR < 60 mL/min/1.73 m^2^ prior to skin scraping excluded from this analysis).

**Table 1 tropicalmed-10-00111-t001:** Association with a skin scraping positive for scabies.

Variable	Total ^a^	Positive	Negative	Odds Ratio ^b^ (95% CI)	*p* ^b^	Adjusted Odds Ratio ^c^ (95% CI)	*p* ^c^
Tested in Cairns	945/4342 (22%)	67/120 (55%)	878/4222 (20%)	4.72 (3.28–6.81)	<0.001	6.18 (4.24–9.02)	<0.001
First Nations Australian	2706/4313 (63%)	110/120 (92%)	2596/4193 (61%)	6.76 (3.53–12.96)	<0.001	10.54 (5.43–20.45)	<0.001
Adult (age > 16)	3555/4345 (82%)	116/121 (96%)	3439/4224 (79%)	5.30 (2.16–13.01)	<0.001	5.94 (2.40–14.69)	<0.001
Female sex	2174/4344 (50%)	70/121 (58%)	2104/4223 (48%)	1.38 (0.96–1.99)	0.08	1.13 (0.77–1.65)	0.53

Percentages rounded to the nearest whole number. ^a^ Incomplete data for some of the 4345 episodes. ^b^ Odds ratio in univariate logistic regression. ^c^ Odds ratio in multivariate logistic regression.

**Table 2 tropicalmed-10-00111-t002:** Characteristics of presentations to Cairns Hospital Emergency Department with a diagnosis of classic scabies and crusted scabies.

	Classic Scabies ^a^ (n = 271)	Crusted Scabies ^a^ (n = 21)	Total (n = 292)
First Nations Australian	197 (73%)	19 (91%)	216 (74.0%)
Male sex	140 (52%)	9 (43%)	149 (51.0%)
Child (<16 years)	150 (55%)	8 (38%)	158 (54.1%)
Scrapings performed	9 (3%)	5 (24%)	14 (4.8%)
Positive skin scrapings	0	1 (5%)	1 (0.3%)
Infectious disease or dermatology service involvement	7 (3%)	6 (29%)	13 (4.5%)
Complete documentation of IACS criteria ^b^	11 (4%)	4 (19%)	15 (5.1%)
Adequate documentation for clinical scabies diagnosis ^b^	115 (42%)	11 (52%)	126 (43.2%)
Satisfied IACS criteria for confirmed scabies ^b^	0	1 (5%)	1/292 (0.3%)
Satisfied IACS for clinical scabies ^b^	88 (33%)	13 (62%)	101 (34.6%)
Documented crusted scabies severity grading	Not Applicable	2 (10%) ^d^	Not Applicable
Followed all NT guidelines for crusted scabies ^c^	Not Applicable	1 (5%)	Not Applicable
Admitted to hospital from the Emergency Department	60 (22%)	16 (76%)	76 (26.0%)

^a^ Based on the attending clinician’s documented assessment. ^b^ IACS criteria for the diagnosis of scabies (Appendix A) [44]. ^c^ Including skin scrapings, dermatology or infectious disease service involvement, assessment of severity, and hospital admission [45]. ^d^ Both cases were seen by dermatology.

## Data Availability

Data cannot be shared publicly because of the Queensland Public Health Act 2005. Data are available from the Far North Queensland Human Research Ethics Committee (contact via email: FNQ_HREC@health.qld.gov.au) for researchers who meet the criteria for access to confidential data.

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
