# Peer review of "The Incidence of Scabies in Far North Queensland, Tropical Australia: Implications for Local Clinical Practice and Public Health Strategies"

_tropicalmed, 2025, doi:10.3390/tropicalmed10040111_

Round 1
Reviewer 1 Report
Comments and Suggestions for Authors
Abstract:
Lacking an objective statement and background on the link to ARF, RHD and CKD.
These abbreviations should not be used in the abstract.
If word count is an issue, it is acceptable not to include so much detail on the results. An OR and HR is sufficient without all the numbers and percentages.
Line 21: make it clear this was a separate study, may be just by adding “ We also reviewed…”
Background
I would suggest adding a few lines of information on ARF, RHD or CKD since these are the underlying reason for this study.
Methods:
Fig 1. It will help non-Australians if you mark the NT which you refer to in the Background and Discussion and also Cairns on the maps. Why are all the shires marked and yet never referred to?
Line 132: When describing the methods and results for ARF and RHD make it clear that the outcome is a combined measure of the number of cases of either. Same for Supplementary Fig 2.
Line 133: perhaps add a sentence to explain dropping cases of ARF, RHD or CKD diagnosed before the skin scraping. Why were they not of interest?
Line 142: I believe this is a typo. representation should be re-presentation.
Results
Table 1. I would normally expect the result of a multivariable logistic regression to be labelled as adjusted Odds ratio (aOR).
Line 223: I believe this is a typo. representation should be re-presentation
Line 224: seems to have a typo
Discussion
Typos line 314 & 332.
Author Response
Reply to Reviewer 1
Response: We thank Reviewer 1 for the time that he/she has taken to review our manuscript and the helpful suggestions that he/she has made for its enhancement. Please find our point-by-point response to his/her comments below.
Abstract:
Lacking an objective statement and background on the link to ARF, RHD and CKD.
These abbreviations should not be used in the abstract.
If word count is an issue, it is acceptable not to include so much detail on the results. An OR and HR is sufficient without all the numbers and percentages.
Response: We thank Reviewer 1 for highlighting this omission. He/she was correct to suggest that the 200-word word count limit for the abstract contributed to the use of abbreviations! We have revised the abstract to address his/her concerns, adding objective statements about ARF and RHD and the association with scabies. We have been able to achieve this within the 200-word word limit by deleting the text describing the component of the study that examined assessment in Emergency department (which was always a secondary component of the study).
Change: Addition of an objective statement and background on the link between scabies and ARF/RHD. All abbreviations in the abstract spelled out at the time of first usage.
Line 21: make it clear this was a separate study, may be just by adding “ We also reviewed…”
Response: We thank Reviewer 1 for the suggestion. We have deleted this section of the abstract now as outlined above.
Change: This text has now been deleted.
Background
I would suggest adding a few lines of information on ARF, RHD or CKD since these are the underlying reason for this study.
Response: We thank Reviewer 1 for highlighting this omission. We have added text to the background as suggested which briefly describes the link between S. pyogenes infection and ARF/RHD and PSGN.
Change: We had added text and references in the introduction that provides background information on the relationship between S. pyogenes infection and ARF/RHD and PSGN and the impact on the health of First Nations Australians.
Methods:
Fig 1. It will help non-Australians if you mark the NT which you refer to in the Background and Discussion and also Cairns on the maps. Why are all the shires marked and yet never referred to?
Response: We agree, and we thank Reviewer 1 for raising this issue. We have revised Figure 1 as suggested.
Change: Revision of figure 1 to highlight the states of Queensland, Western Australia and the Northern Territory. We have highlighted the city of Cairns. We have removed all references to the shires to prevent confusion.
Line 132: When describing the methods and results for ARF and RHD make it clear that the outcome is a combined measure of the number of cases of either. Same for Supplementary Fig 2.
Response: We thank Reviewer 1 for highlighting this omission. We have revised the text accordingly.
Change: We have amended the text to highlight that the outcome is a combined measure of the number of cases of either ARF or RHD.
Line 133: perhaps add a sentence to explain dropping cases of ARF, RHD or CKD diagnosed before the skin scraping. Why were they not of interest?
Response: We thank Reviewer 1 for raising this point. However, as these episodes of ARF/RHD may have been related to S. pyogenes infection that was unrelated to scabies infection (S. pyogenes pharyngitis or, potentially, S. pyogenes infection of the skin that was completely unrelated to scabies) we felt it wiser to drop these cases. The small number of cases of ARF/RHD diagnosed before the skin scraping would have little meaningful impact on our findings.
However, it was notable that individuals who tested positive for scabies were no more likely to have had a prior diagnosis of ARF or RHD compared to matched controls (1/101 (1%) versus 3/101 (3%); p=0.62). Indeed, the absolute number of cases of ARF or RHD was less in the individuals who tested positive for scabies (1 versus 3).
We have been transparent about this, and we feel that the reader will feel that it is reasonable and appropriate that these cases were dropped. We have therefore made no change made to the manuscript.
Line 142: I believe this is a typo. representation should be re-presentation.
Response: We thank Reviewer 1 for highlighting this typographical error. It has been corrected in the revised manuscript (line XXX).
Change: “representation” changed to “re-presentation”.
Results
Table 1. I would normally expect the result of a multivariable logistic regression to be labelled as adjusted Odds ratio (aOR).
Response: We agree with Reviewer 1. We have amended table 1 as suggested.
Change: Addition of “adjusted” to odds ratio for the multivariable logistic regression results
Line 223: I believe this is a typo. representation should be re-presentation
Response: We thank Reviewer 1 for highlighting this typographical error. It has been corrected in the revised manuscript (line XXX)
Change: “representation” changed to “re-presentation”.
Line 224: seems to have a typo
Response: We thank Reviewer 1 for highlighting this typographical error. It has been corrected in the revised manuscript.
Change: “and alternative diagnosis” to “an alternative diagnosis”
Discussion
Typos line 314 & 332.
Response: We thank Reviewer 1 for highlighting these typographical errors. They have been corrected in the revised manuscript.
Change: “believed to at greater risk” to “believed to be at greater risk”
“A public health response to an outbreak of 2021 outbreak of PSGN in Aurukun” changed to “A public health response to a 2021 outbreak of PSGN in Aurukun”
Reviewer 2 Report
Comments and Suggestions for Authors
The manuscript reports the results of a retrospective registry-based epidemiological study of scabies in northern Queensland. The study addresses the spatiotemporal descriptive epidemiology of scabies and assesses whether individuals with confirmed scabies are at higher risk of subsequently developing acute rheumatic fever (ARF), rheumatic heart disease (RHD), and renal impairment (CKD). It also examines the compliance of local clinicians assessing individuals at risk for scabies with contemporary published diagnostic guidelines.
The topic is of some interest, the sample size is large enough for meaningful analysis, and the manuscript is well structured and nicely written. However, some aspects, mostly statistical, need to be clarified.
Main points:
- In Section 3.2, the time trend of the proportion of positive tests is shown in Figure 3 and statistically analyzed by testing whether the Spearman rank correlation coefficient is different from zero. This statistical approach to testing for trend is rather indirect. The Cochran-Armitage test is the standard test for this situation, as it looks directly at whether the time-ordered sequence of proportions is increasing or decreasing monotonically. I am quite sure that the conclusion that the proportions of a positive test are decreasing over time will not change, but the statistical evaluation of this question using the Cochran-Armitage test is more straightforward.
- In Section 3.4, the association between scabies and ARF/RHD/CKD is analyzed based on a nested substudy that included data from confirmed scabies patients and matched controls. From the description of the design of this substudy, it can be concluded that an individual matching procedure was used. In the Statistical Analysis subsection, it is not clear that individual matching was accounted for in the statistical analysis. In particular, clarify whether you used conditional logistic regression modeling and how you handled matching in the Cox analysis.
- Retrospective registry-based studies always have limitations (potential selection bias, incomplete data, etc.) that need to be acknowledged and discussed appropriately. The authors are attentive to the limitations and discuss them appropriately, but I missed (i) a statement about the (probably incomplete) follow-up when addressing the association of scabies and ARF/RHD/CKD (which is - in addition to the mentioned confounding issue - an additional problem) and (ii) a more general discussion about the completeness of the clinical data in the analysis.
Minor points:
- Supplementary Figures 1 to 3 are referred to as "Consort diagrams", which is a rather unusual phrase, at least to my knowledge. The CONSORT statement is a guideline for reporting the results of randomized clinical trials that includes a structured flowchart to visualize the process of enrollment, allocation, follow-up, and analysis of the study population, indicating the size of the study groups at each step. The supplementary figures of this manuscript also visualize the size of the study groups and excluded groups at different stages of the analysis, but the study is a retrospective epidemiological study and not a randomized clinical trial. Therefore, I find it misleading to use the term "consort diagram" in this context and suggest that the term be changed. Alternatively, the authors can refer me to papers (in reputable journals) that have published "Consort diagrams" for retrospective epidemiologic studies (I have only seen "Consort diagrams" for randomized clinical trials).
- The abstract uses a lot of abbreviations (ARF, RHD, IQR) which have not been explained (they are explained in the main text). This should be fixed.
Author Response
Reply to Reviewer 2
Reviewer 2
The manuscript reports the results of a retrospective registry-based epidemiological study of scabies in northern Queensland. The study addresses the spatiotemporal descriptive epidemiology of scabies and assesses whether individuals with confirmed scabies are at higher risk of subsequently developing acute rheumatic fever (ARF), rheumatic heart disease (RHD), and renal impairment (CKD). It also examines the compliance of local clinicians assessing individuals at risk for scabies with contemporary published diagnostic guidelines.
The topic is of some interest, the sample size is large enough for meaningful analysis, and the manuscript is well structured and nicely written. However, some aspects, mostly statistical, need to be clarified.
Response: We thank Reviewer 2 for the time that he/she has taken to review our manuscript and the helpful suggestions that he/she has made for its enhancement. Please find our point-by-point response to his/her comments below.
Main points:
- In Section 3.2, the time trend of the proportion of positive tests is shown in Figure 3 and statistically analyzed by testing whether the Spearman rank correlation coefficient is different from zero. This statistical approach to testing for trend is rather indirect. The Cochran-Armitage test is the standard test for this situation, as it looks directly at whether the time-ordered sequence of proportions is increasing or decreasing monotonically. I am quite sure that the conclusion that the proportions of a positive test are decreasing over time will not change, but the statistical evaluation of this question using the Cochran-Armitage test is more straightforward.
Response: We thank Reviewer 2 for this constructive suggestion. In the revised manuscript we have used the Cochran-Armitage test for this analysis as recommended. The different statistical test does not, as the Reviewer anticipated, change the findings.
- In Section 3.4, the association between scabies and ARF/RHD/CKD is analyzed based on a nested substudy that included data from confirmed scabies patients and matched controls. From the description of the design of this substudy, it can be concluded that an individual matching procedure was used. In the Statistical Analysis subsection, it is not clear that individual matching was accounted for in the statistical analysis. In particular, clarify whether you used conditional logistic regression modeling and how you handled matching in the Cox analysis.
Response: This is an excellent point, and we thank the Reviewer for raising it. The reviewer is correct to suggest that conditional logistic regression modelling is preferred for matched data. The positive cases in our cohort were matched to age, sex, location, and First Nations Australian status (as we state in the methods). Presenting the results of conditional logistic regression modelling that control for all these factors is more complicated and, as we had a relatively small number of cases of ARF or RHD in both the cases and their matched controls, in the revised manuscript we have presented the raw data and analysed the association using Fisher’s exact test. We feel that this illustrates that ARF or RHD is a rare diagnosis in individuals with a scabies (and their matched controls).
We did not control for matching in the Cox analysis, however, we have expanded the limitations section to further emphasise this as a limitation of the paper.
Changes: Dropping of logistic regression analysis of prior ARF/RHD cases; substitution of Fisher’s exact test. Expansion of limitations section to highlight missing data (in addition to the prior text that emphasised that we did not control for confounding factors).
- Retrospective registry-based studies always have limitations (potential selection bias, incomplete data, etc.) that need to be acknowledged and discussed appropriately. The authors are attentive to the limitations and discuss them appropriately, but I missed (i) a statement about the (probably incomplete) follow-up when addressing the association of scabies and ARF/RHD/CKD (which is - in addition to the mentioned confounding issue - an additional problem) and (ii) a more general discussion about the completeness of the clinical data in the analysis.
Response: We agree with Reviewer 2 and thank him/her for highlighting this issue. In the original manuscript we did write “The study’s retrospective design and reliance on documentation in the medical record precluded the collection of comprehensive clinical data.” However, we agree, we could have described the potential for incomplete follow up and resulting incomplete data more specifically.
Change: In the revised manuscript we have added text to the limitations to describe the incomplete clinical and laboratory data and how this may have affected our findings.
Minor points:
- Supplementary Figures 1 to 3 are referred to as "Consort diagrams", which is a rather unusual phrase, at least to my knowledge. The CONSORT statement is a guideline for reporting the results of randomized clinical trials that includes a structured flowchart to visualize the process of enrollment, allocation, follow-up, and analysis of the study population, indicating the size of the study groups at each step. The supplementary figures of this manuscript also visualize the size of the study groups and excluded groups at different stages of the analysis, but the study is a retrospective epidemiological study and not a randomized clinical trial. Therefore, I find it misleading to use the term "consort diagram" in this context and suggest that the term be changed. Alternatively, the authors can refer me to papers (in reputable journals) that have published "Consort diagrams" for retrospective epidemiologic studies (I have only seen "Consort diagrams" for randomized clinical trials).
Response: We thank Reviewer 2 for raising this issue. We have removed the word “consort” from the titles of the figures in every instance where it has been used. We now describe the flowcharts more simply as figures.
- The abstract uses a lot of abbreviations (ARF, RHD, IQR) which have not been explained (they are explained in the main text). This should be fixed.
Response: We thank Reviewer 2 for raising this issue. We have revised the abstract to address this point.
Change: Extensive revision of the abstract. All abbreviations have either been deleted or spelled out at their first time of usage.
Round 2
Reviewer 2 Report
Comments and Suggestions for Authors
The revision has addressed my remarks. Most changes and additions are fine for me.
However, one of my remarks (focusing on the statistical handling of matched data in logistic regression model) has been addressed by eliminating the results of the logistic model and presenting only raw data. The authors state in their reply, "in the revised manuscript we have presented the raw data and analysed the association using Fisher’s exact test". Using Fisher's exact test with individually matched data when comparing a dichotomous outcome between two groups is statistically incorrect as it ignores the matching. Instead, McNemar's test has to be used (due to the sparse data situation the exact version of McNemar's test which is a special form of a binomial test). The authors have to correct this and amend their Statistical Analysis subsection.
A verb is missing in the added part of the sentence in line 420 ("...a quarter of the individuals did not an accessible measure of glomerular filtration rate during follow up".)
Replacing "consort diagram" by another term is fine, but "flowchart" seems to be the better choice than just stating "figure".
Author Response
The revision has addressed my remarks. Most changes and additions are fine for me.
Response: We thank the Reviewer for the time that he/she has taken to review our manuscript and the helpful suggestions that he/she has made for its enhancement. Please find our point-by-point response to his/her comments below.
However, one of my remarks (focusing on the statistical handling of matched data in logistic regression model) has been addressed by eliminating the results of the logistic model and presenting only raw data. The authors state in their reply, "in the revised manuscript we have presented the raw data and analysed the association using Fisher’s exact test". Using Fisher's exact test with individually matched data when comparing a dichotomous outcome between two groups is statistically incorrect as it ignores the matching. Instead, McNemar's test has to be used (due to the sparse data situation the exact version of McNemar's test which is a special form of a binomial test). The authors have to correct this and amend their Statistical Analysis subsection.
Response: We thank the Reviewer for raising this issue. We have repeated our analysis with McNemar’s test rather than Fisher’s text as suggested. This has not changed the study’s findings or conclusions.
Change: Revision of statistical methods section. Use of McNemar’s test rather than Fisher’s exact test to perform the analysis of the matched data.
A verb is missing in the added part of the sentence in line 420 ("...a quarter of the individuals did not an accessible measure of glomerular filtration rate during follow up".)
Response: We thank the Reviewer for highlighting this omission, we have amended the text accordingly.
Replacing "consort diagram" by another term is fine, but "flowchart" seems to be the better choice than just stating "figure".
Response: We thank the Reviewer for raising this issue. We have added “flowchart” as suggested.